# Effects of human mobility on the spread of Dengue in the region of Caldas, Colombia

**Carolina Ospina-Aguirre**[1,2☯]*, **David Soriano-Paños**[3,7☯], **Gerard Olivar-Tost**[4☯], **Cristian C. Galindo-González**[5☯], **Jesús Gómez-Gardeñes**[6,7☯], **Gustavo Osorio**[5☯]

**1** ABCDynamics, Facultad de ciencias exactas y naturales, Universidad Nacional de Colombia - Sede Manizales, Manizales, Colombia, **2** Departamento de electrónica y automatización, Universidad Autonoma de Manizales, Manizales, Colombia, **3** Instituto Gulbenkian de Ciência, Oeiras, Portugal, **4** Departamento de Ciencias Naturales y Tecnología, Universidad de Aysén, Coyhaique, Chile, **5** Percepción y Control Inteligente (PCI), Departamento de Ingeniería Eléctrica, Electrónica y Computación, Universidad Nacional de Colombia - Sede Manizales, Manizales, Colombia, **6** Departamento de Física de la Materia Condensada Facultad de Ciencias, Universidad de Zaragoza, Zaragoza, España, **7** GOTHAM lab, Institute for Biocomputation & Physics of Complex Systems (BIFI), Zaragoza, España

☯ These authors contributed equally to this work.
* cospinaa@unal.edu.co

**Data Availability Statement:** All relevant data are within the manuscript and its Supporting information files.

## Abstract

According to the World Health Organization (WHO), dengue is the most common acute arthropod-borne viral infection in the world. The spread of dengue and other infectious diseases is closely related to human activity and mobility. In this paper we analyze the effect of introducing mobility restrictions as a public health policy on the total number of dengue cases within a population. To perform the analysis, we use a complex metapopulation in which we implement a compartmental propagation model coupled with the mobility of individuals between the patches. This model is used to investigate the spread of dengue in the municipalities of Caldas (CO). Two scenarios corresponding to different types of mobility restrictions are applied. In the first scenario, the effect of restricting mobility is analyzed in three different ways: a) limiting the access to the endemic node but allowing the movement of its inhabitants, b) restricting the diaspora of the inhabitants of the endemic node but allowing the access of outsiders, and c) a total isolation of the inhabitants of the endemic node. In this scenario, the best simulation results are obtained when specific endemic nodes are isolated during a dengue outbreak, obtaining a reduction of up to 2.5% of dengue cases. Finally, the second scenario simulates a total isolation of the network, *i.e.*, mobility between nodes is completely limited. We have found that this control measure increases the number of total dengue cases in the network by 2.36%.

## Author summary

For the World Health Organization, dengue is a disease of public health concern. In recent years there is an increasing trend in the number of dengue cases despite existing prevention and control campaigns. The mobility of the population is considered an important factor in dengue dispersion. In this paper, we are interested in addressing how

**Funding:** COA acknowledges the funding provided by Colciencias through a doctoral scholarship under the 'national doctoral scholarships call 647 of 2014,' which has been essential in supporting her doctoral studies and contributing to the progress of this research. JGG acknowledges financial support from the Departamento de Industria e Innovación del Gobierno de Aragón y Fondo Social Europeo (FENOL group grant E36-23R) and from MICIN through project PID2020-113582GB-I00/AEI/10.13039/501100011033. The funders had no role in the study design, data collection and analysis, decision to publish, or manuscript preparation.

**Competing interests:** The authors have declared that no competing interests exist.

restrictions to human mobility might affect the incidence of dengue in a region. Our research is relevant because the model can be adapted to other regions or scales, and the mobility control measures can be taken into account for the development of public health policies in endemic regions.

# 1 Introduction

According to the World Health Organization (WHO), dengue fever is the acute arthropod-borne viral infection with the highest incidence in humans in the world [1]. It is transmitted to humans mainly by the bite of a mosquito *Aedes aegypti* infected with dengue virus (DENV). As a vector-borne disease the transmission between humans only occurs through the bite of infected mosquitoes, never from one person to another. In other words, the mosquito does not cause the disease directly, but acts as a bridge between two people, one with the virus and the other without it. In turn, the mosquito becomes infected when it feeds on the blood of a person infected with dengue and transmits the virus when it bites healthy people.

Mathematical models are extremely useful to understand mechanisms that drive a healthy population toward an epidemic or endemic state, as well as for evaluating containment measures that help to suppress, or at least mitigate, the incidence of a given communicable disease. The usual mathematical approach to such tasks is the use of compartmental models. In these models, individuals in a population can be divided into classes or compartments according to their epidemiological state. For instance, in the celebrated Susceptible-Infectious-Recovered (SIR) model, individuals are divided into susceptible (healthy people who may acquire the virus), infectious (people who have acquired and can transmit the virus) and recovered (people who cannot propagate the pathogen and have acquired immunity to the virus) [2–4]. Yet simple and minimal, the SIR model has been pervasively used to analyse a plethora of viral infections such as measles [5, 6], rubella [7], malaria [8], zika [9], COVID-19 [10, 11], dengue [12–18], among others.

In the case of vector-borne diseases, more refined compartmental models have been introduced in which both human and vector populations are divided into several compartments. For instance in [19, 20] dynamics of human population is modeled through a SIR model whereas vectors are divided into Susceptible and Infectious. This division creates a SIR-SI model which is the compartmental dynamics adopted in our work. Alternative approaches to the study of DENV transmission include the Ross-Macdonald epidemic model [21] where the gradual acquisition of immunity in humans is ignored or the addition of further compartments to capture the growth dynamics of the vector population [22, 23]. In all these approaches, the main goal is to define control strategies that favor the eradication of the virus in human populations.

Most of the control strategies rely on improving hygiene measures, the use of pesticides or, more recently, the release of Wolbachia-infected vectors in high incidence habitats [23]. However, human behavior plays an important role in the geographical spread of pathogens [24, 25], and vector-borne diseases are not the exception. One of the most salient features of human behavior affecting the spread of vector-borne diseases is human mobility [26] which, in the case of dengue, has become an essential factor given the expansion of urban environments and the increased frequency of international travels [27–29]. For the particular case of dengue transmission, mobility determines the degree of exposure to vectors [30, 31]. Therefore, it is essential to avoid local dengue outbreaks in low incidence areas through the importation of cases from distant and high incidence regions [32, 33]. A remarkable recent example of

such phenomenon was the reintroduction of the dengue virus in Singapore [34] where dengue was successfully controlled, although the effects were not sustainable in the long term given the large movement of people and trade material in the dengue-endemic region. The contribution of human mobility to the spread of dengue cases has become an essential factor given the increasing frequency of international travel and the expansion of urban environments [27, 28]), since, in the vast majority of cases, it is people who create artificial breeding sites for the vector that spreads the disease [26, 35]. It is projected that by 2030, over half of the world's population will reside in urban areas of the tropics, due to population growth and migration from rural areas [36].

The importance of human behavior and, in particular, human mobility in the spread of communicable diseases has motivated the formulation of different mathematical frameworks to study the contribution of mobility in the spread of vector-borne diseases [12, 37, 38]. Most of these approaches bridge the gap between single-population models to metapopulations by incorporating the complex architecture of human flows in the form of networks [39] defined through data-driven frameworks [40–42]. Equipped with this framework, we aim to analyze how mobility constraints affect dengue transmission using two control scenarios. In the first scenario, we focus on limiting mobility in areas with the highest dengue incidence. As for the second scenario, we explore three types of mobility restrictions in a fully connected network. The first one is to allow residents to leave a municipality but prevent outsiders from entering. Another measure involves allowing outsiders to enter a municipality with a high incidence of dengue but restricting the departure of its residents. Lastly, we implemented isolation by preventing both residents from leaving and outsiders from entering a municipality. In the second scenario, the effect of confinement is analyzed, which involves completely restricting mobility throughout the network, prohibiting visitors from entering and residents from leaving a municipality. Surprisingly, with isolation applied, there is an increase of 2.36% in dengue cases compared to the unrestricted scenario.

This article is organized as follows. Section 2, Materials and methods, describes the proposed compartmental model, the methodology used for estimating the model parameters and the coupling of the epidemiological model with the mobility network. Section 3 presents the results of the case study in which the model was applied in the department of Caldas—Colombia, where we analyze the effectiveness of the different mobility restrictions described above. The manuscript concludes in section 4 with a brief discussion of the results and their implication for informing public health decision makers.

## 2 Materials and methods

According to Pastor-Satorras and collaborators [39], we are currently witnessing a golden age in epidemic modeling: models are improving significantly driven by the continuous addition of data, while the powerful computational resources available now make it possible to extend simulations to new limits. Throughout the years, traditional compartmental models such as the SIR one have been refined and completed to analyse the transmission of real communicable diseases such as measles [5, 6], rubella [7], malaria [8], zika [9], COVID-19 [10, 11], dengue [12–18, 38] and others. In the following, we present our dynamical framework: a SIR-SI epidemic model coupled with human mobility.

### 2.1 SIR-SI compartmental model

The compartmental dynamics governs the contagion and recovery processes at the level of a single region (or a patch in the metapopulation framework) in which humans and mosquitoes (the vectors) coexist. Let us denote the total size of the populations of humans and mosquitoes

by $N_h$ and $N_m$, respectively. Human population is divided into susceptible ($S_h$), infectious ($I_h$) and recovered ($R_h$), while mosquitoes population is divided into susceptible ($S_m$) and infectious ($I_m$) ones. To define the compartmental dynamics, the following assumptions are made:

- Incubation period is neglected for both humans and mosquitoes.

- Deaths caused by the disease are not considered for either humans or mosquitoes [21].

- Mosquitoes cannot recover after being infected [21].

- There is not super-infection for either humans or mosquitoes [21].

- Susceptible humans can only get infected trough an infected mosquito bite.

- Human recruitment rate is constant [21].

- Mosquito recruitment rate depends on environmental conditions and is constant in each region [21].

Most of the former assumptions follow those typically used when studying the dynamics of dengue transmission and potential control strategies, as Sepulveda [21] did in her study conducted in Cali, Colombia. Following her work one can derive (explained below) a set of differential equations that serve as a fundamental framework to represent the interactions between susceptible, infected, and recovered individuals in a population, along with the transmission dynamics involving the mosquito vector. The rest of the assumptions, which encompass aspects related to population dynamics and the compartmental model, are commonly made when studying the spread of vector-borne diseases, enabling the exploration of different scenarios and control strategies [43].

Once reported the assumptions of the model, let us derive the basic equations driving the time evolution of the compartments associated to both humans and vectors. This way, a susceptible mosquito can become infectious if it comes into contact with an infected individual. The likelihood of this transmission is influenced by several factors, including the transmission probability for the vector, denoted by $\lambda_m$, the bite rate $\beta$, and the proportion of infectious humans in the region, represented as $\frac{I_h}{N_h}$. We can express the infection rate for susceptible mosquitoes as $\frac{\beta \lambda_m I_h}{N_h}$. Similarly, for humans, the infection rate depends on the biting rate $\beta$, the number of infectious mosquitoes $I_m$, the transmission probability $\lambda_h$, and is inversely proportional to the total human population $N_h$. This rate determines the probability of a specific individual being bitten and infected. We can represent the infection rate for susceptible humans as $\frac{\beta \lambda_h I_m}{N_h}$. Considering the former mechanisms, the dynamical evolution of each compartment associated to a given municipality is described by the following differential equations:

$$
\begin{aligned}
\dot{S}_m &= \Lambda_m - \frac{\beta \lambda_m I_h}{N_h} S_m - \delta_m S_m \ , \\
\dot{I}_m &= \frac{\beta \lambda_m I_h}{N_h} S_m - \delta_m I_m \ , \\
\dot{S}_h &= \Lambda_h - \frac{\beta \lambda_h I_m}{N_h} S_h - \delta_h S_h \ , \\
\dot{I}_h &= \frac{\beta \lambda_h I_m}{N_h} S_h - \mu I_h - \delta_h I_h \ , \\
\dot{R}_h &= \mu I_h - \delta_h R_h \ ,
\end{aligned}
\tag{1}
$$

**Table 1. Parameters description for the SIR-SI model.**

| Parameter | Description |
|-----------|-------------|
| $\beta$ | Average bite rate per time unit of the mosquito. |
| $\lambda_h$ | Transmission probability from infectious mosquitoes to susceptible humans per bite. |
| $\lambda_m$ | Transmission probability from infectious humans to susceptible mosquitoes per bite. |
| $\delta_m$ | Natural death rate for mosquitoes. |
| $\mu$ | Recovery rate for humans. |
| $\delta_h$ | Natural death rate for humans. |
| $\Lambda_m$ | Recruitment rate of susceptible mosquitoes. |
| $\Lambda_h$ | Recruitment rate of susceptible humans. |

with $N_h = S_h + I_h + R_h$. In Table 1 we specify the definition of the parameters used in the former equations of the SIR-SI model.

The dynamics of the infectious set in the SIR model according to Allen [44] can yield two different behaviors, one in which there is an epidemic peak and the other in which the number of infected individuals decreases monotonically towards 0. In [45] such analysis is extended to the SIR-SI model (1) and it is shown that the dynamics for the infected population depends on:

$$\rho = \sqrt{\frac{\beta^2 \lambda_m \lambda_h}{\delta_m(\delta_h + \mu)} \frac{N_m^*}{N_h^*}} = \sqrt{\frac{\beta^2 \lambda_m \lambda_h}{\delta_m(\delta_h + \mu)} \frac{\Lambda_m \delta_h}{\Lambda_h \delta_m}} \tag{2}$$

When $\rho > 1$, the dynamics of infected individuals shows a maximum (epidemic peak) while when $\rho < 1$ the number of infectious agents display a monotonous decreasing behavior and thus no outbreak occurs.

**2.1.1 Parameter estimation and initialization.** The parameters governing contagion dynamics and the mosquito's life-cycle are extracted from the work published by Helmersson *et al.* [46]. In this manuscript, the authors proposed different equations capturing the reduction of vectorial capacity of *Aedes aegypti* when being exposed to either very high ($T > 34°C$) or very low ($T < 12°C$) temperatures. The behavior of the parameters as a function of temperature can be seen Fig 1. In particular, the average bite rate reads:

$$\beta(T) = 0.0043T + 0.00943 \quad (day^{-1}) \quad 12.4 \le T \le 32 , \tag{3}$$

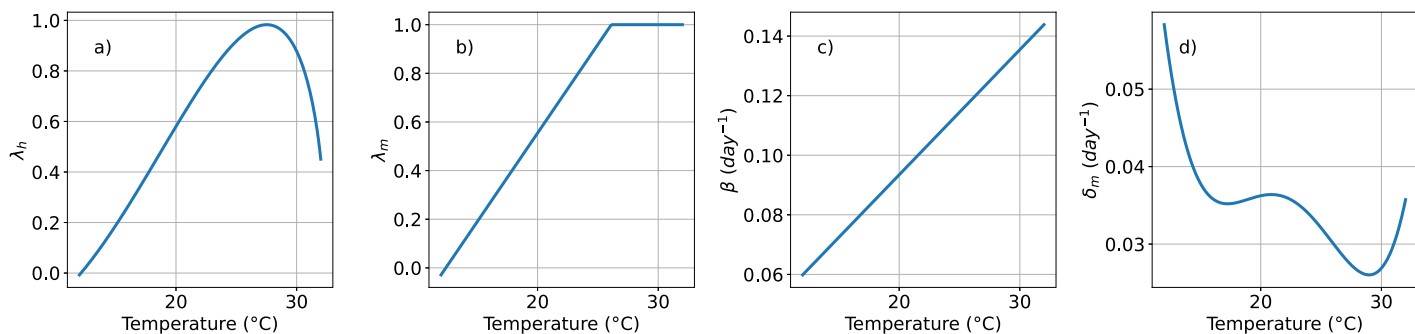

**Fig 1. Temperature dependence of the parameters.** (a) Likelihood of a person becoming infectious $\lambda_h$, (b) Likelihood of a mosquito becoming infectious $\lambda_m$, (c) Bite rate $\beta$, (d) Rate of death of mosquitoes $\delta_m$.

while the transmission probability from infectious humans to susceptible mosquitoes per bite is given by:

$$\lambda_m(T) = \begin{cases} 0.0729T - 0.9037 & 12.4 \leq T \leq 26.1 \ , \\ 1 & 26.1 < T \leq 32.5 \ . \end{cases} \qquad (4)$$

and, in the range $12.286 \leq T \leq 32.461$, the transmission probability per bite from infectious mosquitoes to susceptible humans can be modeled as:

$$\lambda_h(T) = 0.001044T(T - 12.286)\sqrt{32.461 - T} \ . \qquad (5)$$

Finally, there is a fourth parameter that depends on temperature, the mosquito death rate, that can be expressed as:

$$\delta_m(T) = 0.8692 - 0.1590T + 0.01116T^2 - 3.408 * 10^{-4}T^3 + 3.809 * 10^{-6}T^4 \ , \qquad (6)$$

for $10.54 \leq T \leq 33.41$. In Fig 1 we show the variation of these four parameters as the temperature increases from 12°C to 32°C.

The remaining four parameters of the SIR-SI model are defined as follows. For $\Lambda_h$, the approach proposed in [47] is used, where it is defined as $\Lambda_h = \frac{N_h}{EVH * 365}$. The recovery rate for humans is set to $\mu = 0.32288 \ day^{-1}$ according to the study developed in [14], while $\delta_h$ is estimated using known specific statistics from the geographical region under analysis. Finally, to estimate $\Lambda_m$ we start from the equation for the vital dynamics of the vector when $S_m = 0$, (i.e. $\dot{S}_m = \Lambda_m - \delta_m$), where the carrying capacity (number of mosquitoes in the steady state) is obtained as $S_m^* = \frac{\Lambda_m}{\delta_m}$. The recruitment rate, therefore, can be expressed in terms of the carrying capacity $\Lambda_m = S_m^* \delta_m$.

Once explained the estimation of the model parameters, we now focus on the initialization of the model. To set the initial conditions for the model, some considerations have to be mentioned. First of all, it has to be noted that $N_h$ can be fixed based on the known population of each municipality, while $N_m$ can be estimated with the Index of Domiciliary Infestation (number of houses with one or more containers positive for immature *Aedes aegypti* divided by the number of houses sampled multiplied by 100) and the average number of people per household in the study region.

Considering Eq (1) and the parameterization described above, we present in Fig 2 the behavior of the model while simulating a region with an average temperature of 31°C (which yields $\beta = 0.227$, $\lambda_m = 1$, $\delta_m = 0.0298$, $\lambda_h = 0.732$, and $\mu = 0.329$) and a mosquito recruitment rate of $\Lambda_m = 86.634$. For the first set of initial conditions ($N_m = 148.352$, $S_m = 118.352$, $I_m = 30$, $N_h = 76963$, $S_h = 76963$, $I_h = 0$), correspond to $\rho > 1$, and thus the dynamics of the model display the expected epidemic peak. In contrast, when using the second set of initial conditions ($N_m = 148.352$, $S_m = 118.352$, $I_m = 0$, $N_h = 76963$, $S_h = 76928$, $I_h = 35$) one obtains $\rho < 1$ and, consequently, the simulation results shows that the number of infected individuals decreases to zero over time.

Finally, let us remark some aspects regarding the parameter fitting process and the simulation of the model. First, we chose a 365 days period for the simulations because we had access to yearly dengue case records. This allowed us to train our model using a gradient descent method, in which the cost function was the error between the simulated one-year incidence of dengue cases and the actual reported cases from local health authorities. In addition, by using a daily time scale, we were able to capture the finer dynamics of dengue transmission and accurately represent the temporal patterns of the disease. The annual average temperature is used

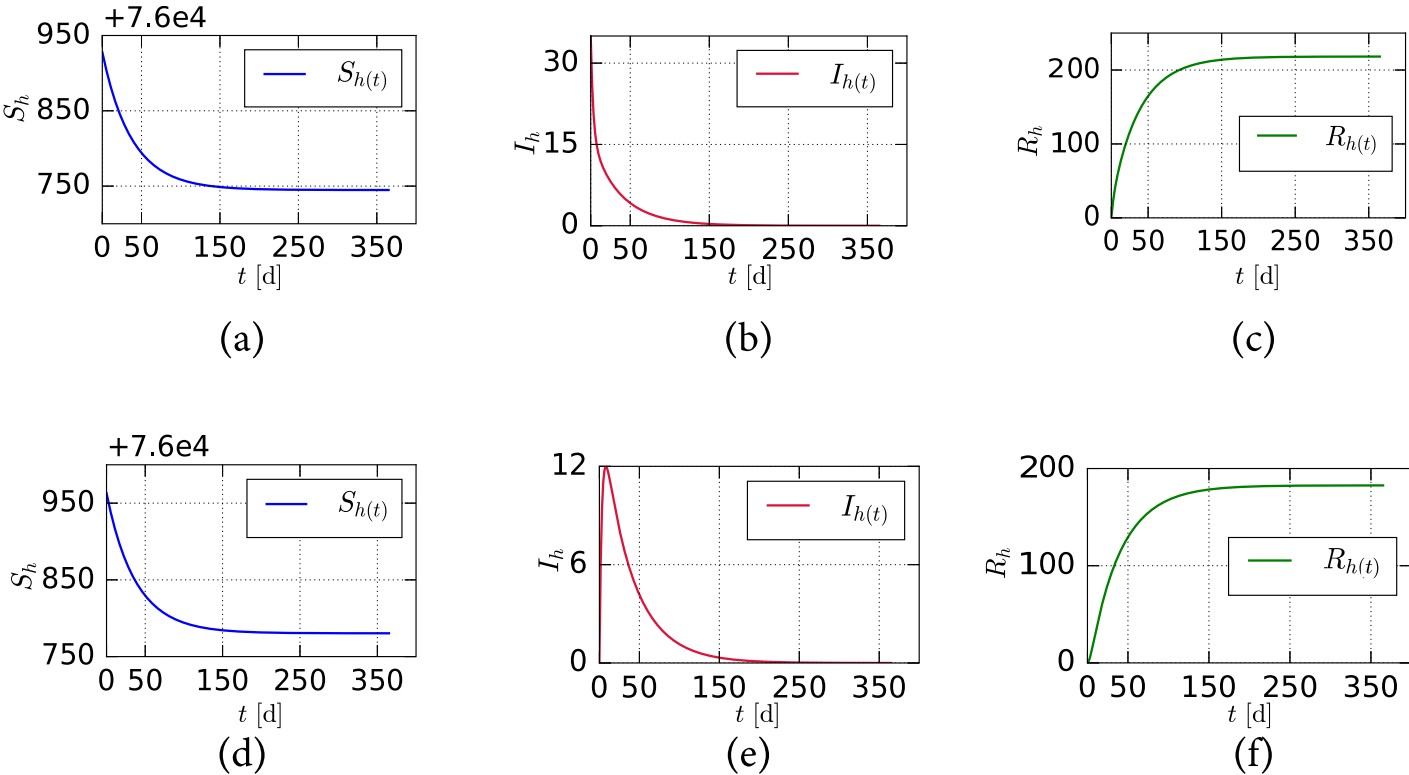

**Fig 2. Behavior of the human population over time in the SIR-SI model.** (a,c) Susceptible, (b,e) Infected and (c,f) Recovered. When $\rho < 1$ (a, b and c), and $\rho > 1$ (d,e and f).

for fitting a model to a specific locality, and once the parameters are calculated using the procedure described in this section, they remain constant throughout the simulation time. This approach improved the clarity and accuracy of our model.

## 2.2 SIR-SI model with human mobility

Once we have introduced the basic ingredients of the SIR-SI model, we turn our attention to its application in a metapopulation framework that incorporates recurrent human mobility patterns. In this context, individuals reside in specific patches but have the freedom to move to other destinations, such as workplaces. In the following we adopt a continous-time version of the framework introduced by Soriano-Paños *et al* [38, 48, 49] as it allows to capture the recurrent nature of human commuting flows. We assume that each area $i$ is characterized by its number of residents $N_h^i$ and its number of mosquitoes $N_m^i$.

To represent the mobility flows, we introduce a matrix $\Upsilon$ in which each element, $\Upsilon_{ij}$, denotes the fraction of the population living in patch $i$ and traveling to patch $j$. Due to a lack of specific data, we assume these movements to follow a fully-connected weighted matrix $\Upsilon$. To construct this matrix, we synthetically assume that 90% of the population within each patch stays there, while 10% of the population moves to other areas. The assumption of 10% is based on the average number of daily travelers between the different municipalities in the Department of Caldas, as derived from information gathered at municipal terminals. We compute the flows connecting different patches following the gravity model [41], implying that the number of connections between two patches $i$ and $j$, hereinafter denoted by $W_{ij}$, can be

expressed as:

$$W_{ij} = \frac{N_h^i N_h^j}{d_{ij}^2} \quad . \tag{7}$$

Taking into account the flows distribution, the elements of the mobility matrix $\Upsilon$ read:

$$\Upsilon = \frac{W_{ij}}{\sum_{l=1}^{n} W_{il}} \quad . \tag{8}$$

Agents' movements between patches $i$ and $j$ cause a redistribution of the population across the system; therefore it is necessary to adjust the equations (1) to account for the actual number of people that are in any patch $i$ at any given time. In particular, the effective population ($N_{he}$) of patch $i$ is defined as:

$$N_{he}^i = \sum_{j=1}^{n} \Upsilon_{ji} N_h^j \quad , \tag{9}$$

which accounts for the distribution of the residential population $\vec{N}$ and the mobility patterns of the individuals of the metapopulation $\Upsilon$. Likewise, mobility also changes the effective number of infected ($I_{he}$) individuals in each patch $i$, which now reads:

$$I_{he}^i = \sum_{j=1}^{n} \Upsilon_{ji} I_h^j \quad , \tag{10}$$

where $I_h^j$ stands for the number of infected individuals living in patch $j$. Finally, we assume that mosquitoes stay inside their associated area. To model the spatio-temporal evolution of the disease, we define the quantity $x^i(t)$ as the occupation of each compartment ($x \in \{S_h, I_h, R_h, S_m, I_m\}$) inside each patch $i$. Following the assumptions of the model, the time evolution of these quantities is given by:

$$
\begin{aligned}
\dot{S}_m^i &= \Lambda_m^i - \frac{\beta^i \lambda_m^i I_{he}^i}{N_{he}^i} S_m^i - \delta_m^i S_m^i \\[6pt]
\dot{I}_m^i &= \frac{\beta^i \lambda_m^i I_{he}^i}{N_{he}^i} S_m^i - \delta_m^i I_m^i \\[6pt]
\dot{S}_h^i &= \Lambda_h^i - S_h^i \sum_{j=1}^{n} \beta^j \lambda_h^j \Upsilon_{ij} \frac{I_m^j}{N_{he}^j} - \delta_h^i S_h^i \\[6pt]
\dot{I}_h^i &= S_h^i \sum_{j=1}^{n} \beta^j \lambda_h^j \Upsilon_{ij} \frac{I_m^j}{N_{he}^j} - (\mu + \delta_h^i) I_h^i \\[6pt]
\dot{R}_h^i &= \mu I_h^i - \delta_h^i R_h^i
\end{aligned}
\tag{11}
$$

In Fig 3 we show an scheme of the metapopulation model in which the SIR-SI compartmental model is integrated with human mobility flows. Let us finally note that the mobility restrictions implemented in this work consisted of setting the rows or columns of the matrix **W** corresponding to the patch under mobility controls to zero. By assigning a value of zero to the entries in the matrix that represent mobility rates between the affected patch, the population flow between these nodes is prevented, effectively restricting their mobility completely.

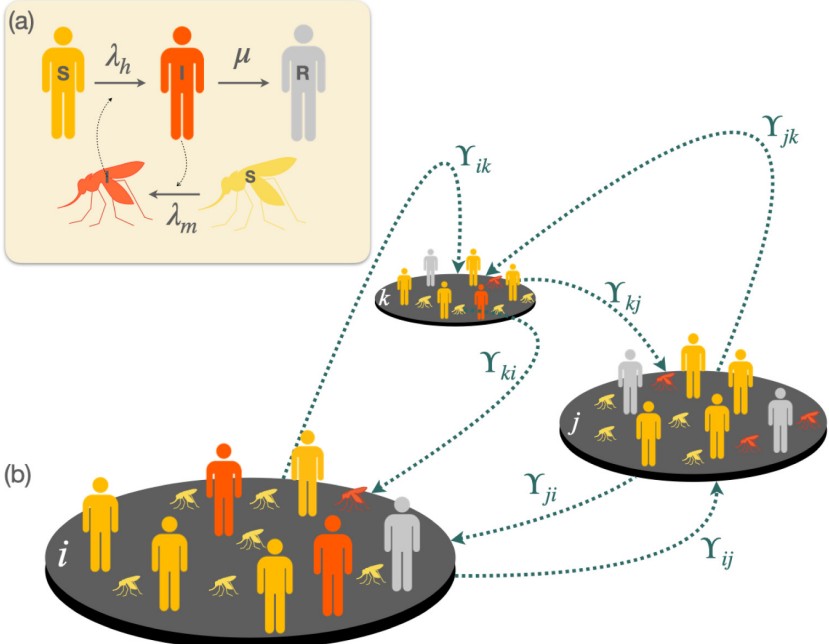

**Fig 3. Schematic view of the SIR-SI metapopulation model.** (a) shows the compartmental dynamics of the SIR-SI model with cross-infections between vectors and humans while panel (b) shows a toy metapopulation of 3 patches connected through links whose weights are given by matrix $\Upsilon$.

## Results

We begin by presenting the general formalism for studying the dissemination of dengue fever in a metapopulation. Next, we explore three simple cases that highlight how mobility between a small set of patches can significantly influence the overall extent of dengue cases. Subsequently, we delve into our primary case study, focusing on the spread of dengue fever in the department of Caldas (CO). We discuss the department's key attributes, how they fit into the metapopulation framework, and how we parameterize the model based on observed dengue cases. Finally, the results of two different mobility restriction scenarios are presented, showcasing their respective impacts on the total number of dengue fever cases.

### 3.1 Mobility effects over dengue cases in simple metapopulations

Starting with an investigation into the impact of mobility on dengue spread, we analyze three straightforward metapopulations. These scenarios serve to demonstrate the effect of varying distances between nodes, with increasing distance leading to reduced mobility according to the gravity model. The three case examples are illustrated in Fig 4.

**3.1.1 Case A.** In Fig 4(a), we present a case example representing a patch (node 1) with zero incidence connected to an endemic patch (node 2). The model parameters utilized for simulating this example can be found in Table A in S1 Text. The evolution of dengue cases in both patches is illustrated in Fig 5a as the distance between them increases. Notably, increasing the distance (and thus reducing mobility) between patches results in an overall increase in the total number of cases. However, it is crucial to recognize that the effects of increasing distance (reducing mobility) for both patches are opposite. Specifically, while increased mobility enhances the exposure of individuals from non-endemic areas, leading to a higher incidence,

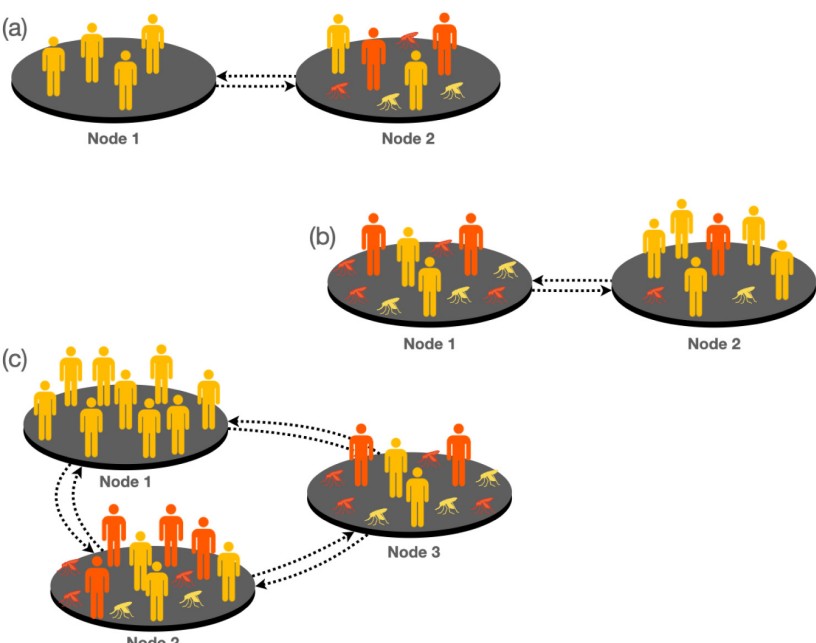

**Fig 4. Schematic plot of the three simple metapopulations covered as case examples.** In panel (a), corresponding to case A, node 1 is a zero-incidence node, i.e., there are no mosquitoes on it, however in node 2, there are environmental conditions for vector breeding and the disease is present. Panel (b) corresponds to case B. In this case node 1 is defined as endemic and is connected to a non-endemic node. Note that in both nodes the disease is present, but in node 1 the vector breeding conditions are better, so there is a wider spread of the disease than in node 2. Finally in panel (c) we define case C for which there are three equidistant connected nodes. Node 1 has zero incidence and its population is the largest while nodes 2 and 3 have characteristics of endemic regions. However, node 2 has more inhabitants than node 3 and, therefore, the probability of infection in node 3 is larger than in the other two patches.

movement across patches benefits the population in endemic areas by decreasing the total incidence of the virus within this population.

**3.1.2 Case B.** In this example, we consider two nodes: node 1 represents a patch with a high disease presence, while node 2 is a patch with fewer dengue cases (refer to Table B in S1 Text for the parameters used). Examining Fig 5b, we observe that reducing mobility in the

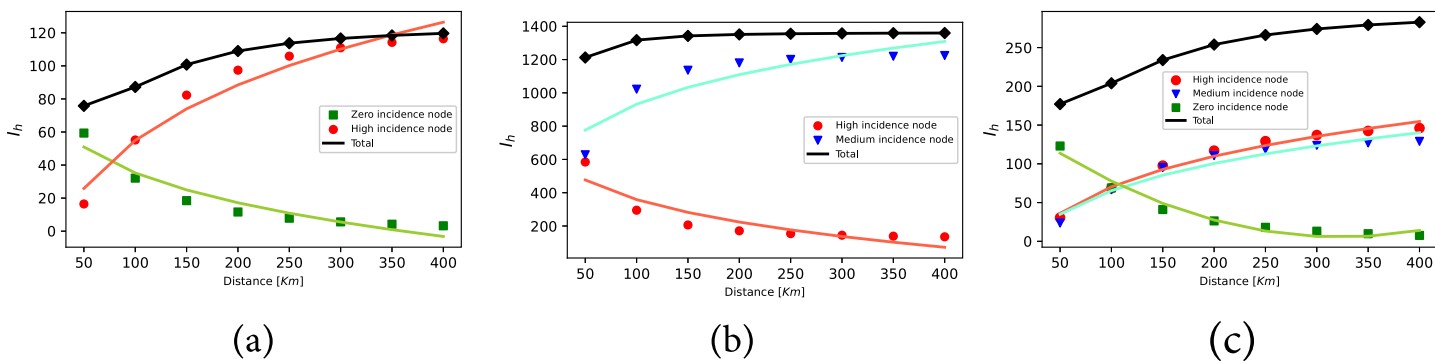

**Fig 5.** Evolution of dengue cases in patches as the distance increases when connecting a) High and a zero incidence node. b) High and medium incidence node. c) High, medium and zero incidence node. The green dots show the dengue cases in the zero incidence node, the blue dots show the cases in the medium incidence node and the red dots show the cases in the high incidence node. The black dots are used for the total number of cases. Lines are added to guide the reader's eye.

endemic node still results in an increase in dengue cases. Conversely, in the node with lower incidence, reducing mobility has a positive effect, leading to a reduction of cases.

**3.1.3 Case C.** In this example, three equidistant nodes are interconnected. Node 1, with the largest population, shows zero incidence according to the parameters indicated in Table C in S1 Text. On the other hand, node 2 has a larger population than node 3, but the latter exhibits a higher disease incidence when isolated. Fig 5c illustrates the results of this scenario. Once again, an increase in the distance between patches, *i.e.*, geographical isolation, leads to an overall increase of dengue cases. However, locally, the situation for node 3 (zero incidence) worsens when the distance is reduced, while for nodes with medium and high incidence, proximity reduces the number of dengue cases by up to 19%. The findings clearly demonstrate that mobility restrictions have a significant impact on reducing the number of cases, especially in areas with a higher incidence of the disease. Nevertheless, it is essential to acknowledge that locally, in some areas, the epidemiological situation may worsen as a byproduct of these containment measures. Motivated by these results, we now propose scenarios to evaluate the effect of applying mobility restrictions in a more complex metapopulation model inspired by the municipalities in the department of Caldas (CO).

## 3.2 Defining the metapopulation of the department of Caldas

The department of Caldas, situated in the central west of the Colombian Andean region, features a mountainous topography, intersected by the Central and Western mountain ranges. With an area of 7888 $km^2$ and a population of 987991 inhabitants [50], its elevation ranges from 5400 masl to 170 masl. The department comprises 27 municipalities, among which ten have direct connections with six other municipalities in neighboring departments. To model the spread of dengue in Caldas using the SIR-SI model, we construct a metapopulation with 33 patches, each representing a municipality and reflecting its census-reported population.

The interconnections between municipalities are determined based on road distances, forming the $\Upsilon$ matrix that governs the transition rates between pairs of municipalities. Notably, this matrix does not distinguish between municipalities within Caldas and those from neighboring departments that share close proximity and interconnectedness. The specific transition rates for all 33 municipalities are detailed in Table D in S1 Text. The procedure employed to compute the $\Upsilon$ matrix can be found in Section 2.2. Population data utilized for these calculations were sourced from authoritative records of the DANE, while distances between municipalities were obtained through Google Maps.

For the numerical simulations, we simplified the model by considering only one dengue serotype, assuming that individuals can be infected once. The key parameters, $\beta$, $\lambda_h$, $\lambda_m$, and $\delta_m$, were individually calculated for each municipality using the method outlined in Section 2.1.1 and are shown in Table E in S1 Text. Let us note that in these estimations we have used the average temperature of each municipality, shown in Table F in S1 Text. Although one can use the time series of the temperatures recorded along the year, the slight variations of it in the region of Caldas allow us to make such simplification.

To account for the impact of the rainy seasons in Caldas, which occur twice a year, we introduced periodic perturbations to our model. These perturbations represent an increase in the number of infected mosquitoes during the two rainy seasons, modifying the values of infected mosquitoes for each municipality from $I_{mDrynees}$ to $I_{mRain}$ (see Table G in S1 Text). This calibration was aimed at capturing the historical dengue case numbers during outbreaks in the department of Caldas (CO) over the past decade, considering the well-established relationship between rainfall and dengue fever cases [35, 46, 51–54].

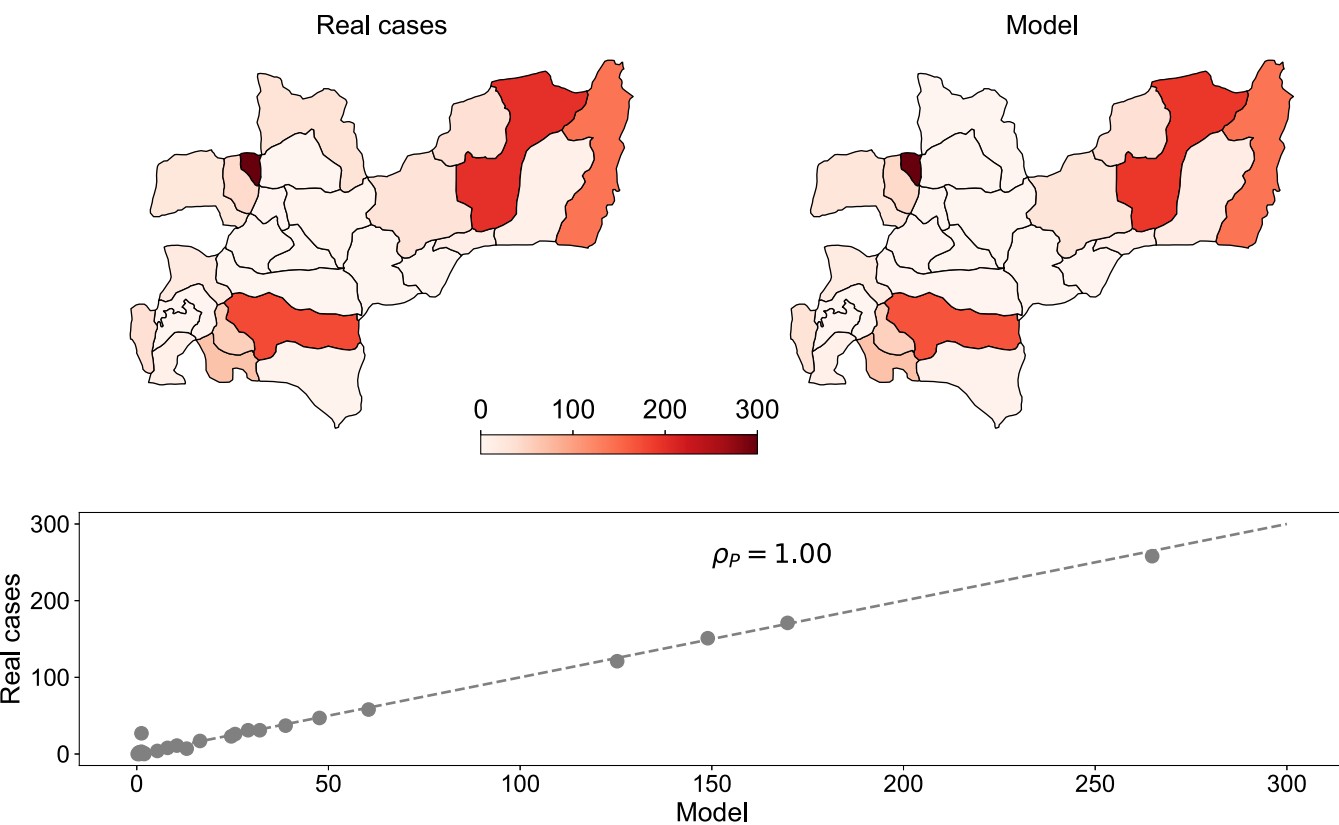

**Fig 6.** Top row: Spatial distribution of the cases of Dengue reported in the department of Caldas in 2015 (Left panel) and that predicted by our model (Right panel). Bottom row: Cases predicted by the model as a function of the reported data in 2015 for each municipality in the department of Caldas. $\rho_P$ indicates the Pearson correlation coefficient between these variables. The dashed grey line represent the perfect agreement scenario between model and real data. Source of the Base Layer: The base layer data used in this map was obtained from Health Observatory of Caldas through ArcGIS Open Data at Gobernación de Caldas website.

The recruitment rate for humans was determined using the life expectancy value ($EVH = 74.64$) from public health records, while the mortality rate for humans was obtained from DANE statistics. Through model adjustment, we minimized the error between simulated and real cases. The calibrated SIR-SI simulation model, with all 33 municipalities interconnected, resulted in 1033 dengue cases in Caldas. The spatial distribution of cases predicted by the model can be found in Table H in S1 Text.

To validate the model, we conducted a comparison between the spatial distribution of reported cases in the department of Caldas during 2015 and the predictions obtained from our equations after calibrating the parameters (see Fig 6). The model's predictions exhibit a strong positive correlation with the actual reported cases, yielding a Pearson correlation coefficient of $\rho_P \simeq 1$. This high correlation demonstrates that the model effectively reproduces the spatial distribution of dengue cases in the department of Caldas during 2015. The specific data used for calibration can be found in Table I in S1 Text.

### 3.3 Control scenario 1: Mobility restrictions focused on high incidence nodes

The main purpose of this scenario is to analyze the impact of mobility constraints on the total number of dengue cases in the network. We investigate the effects of three control measures: i)

restricting access (no foreigners allowed to enter the municipality), ii) restricting departure (inhabitants cannot leave the municipality), and iii) complete isolation (no entry or exit allowed for residents or visitors). These measures were applied in simulations for the four municipalities with the highest number of dengue cases between 2015 and 2019, as reported by the public health office of Caldas [55]. The municipalities considered are La Dorada, Norcasia, Marmato, and Chinchiná. Additionally, the same control measures were implemented in Manizales, the capital city, which experiences significant mobility from the other municipalities.

Fig 7 illustrates the variations in the total number of dengue cases in the department of Caldas when the proposed mobility control measures are applied to the five mentioned municipalities. The graph clearly shows that as mobility in each municipality is reduced, the total number of dengue cases in the department also varies.

**3.3.1 Access.** The control measure is applied to each of the five municipalities, both individually and collectively. When access to La Dorada is limited, a reduction in the total number of cases in the department is observed. In contrast, applying the same measure in Norcasia, Marmato, and Chinchiná results in an increase of the total cases. Additionally, reducing access to Manizales does not lead to any variation in the total number of cases. However, when access is collectively restricted in all five municipalities, there is an increase in the total number of dengue cases in the department (see Fig 7). These results are presented in Table J in S1 Text.

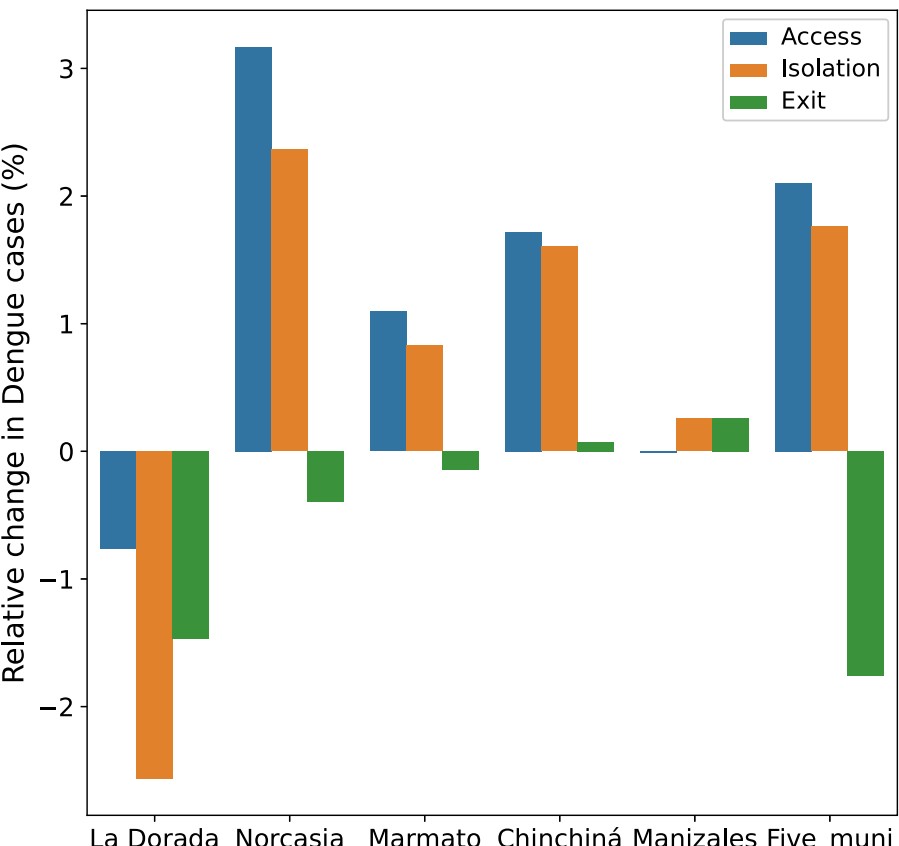

**Fig 7. Variation of dengue cases in the department Caldas according to mobility restrictions when: i) no foreigners are allowed to enter the municipality (blue), no inhabitants can leave the municipality (green), the municipality is totally isolated (orange).**

The impact of mobility restrictions on dengue cases can vary significantly depending on the specific municipality, and it highlights the importance of considering the unique characteristics and dynamics of each region when implementing control measures.

**3.3.2 Exit.** When the departure of residents from the municipality of La Dorada is restricted while allowing the entry of outsiders, the number of dengue cases in that municipality decreases by 52 cases. Conversely, in Norcasia, which exhibits the highest mobility towards La Dorada according to Table D in S1 Text, there is an increase of 31 cases. As a result of this mobility restriction to La Dorada, the total number of dengue cases in the department is reduced by 15 compared to the scenario of unrestricted mobility. In the case of Norcasia and Marmato, when the restriction of departure is implemented, there is an increase of 4 and 2 dengue cases, respectively, of the total number of cases in the department. Please refer to Table J in S1 Text for specific details. On the other hand, when this control measure is applied in Chinchiná, there is a reduction in the number of cases in this municipality. However, the situation is different for Palestina and Manizales. In the case of Palestina there is an increase of four cases while Manizales (the municipality to which the majority of Chinchiná residents travel as shown in Table D in S1 Text) experiences an increase of one dengue case when the mobility restriction is implemented. These results indicate that the mobility patterns between Chinchiná and these two municipalities play a significant role in the observed changes in dengue cases. Consequently, the total number of dengue cases in the department increases by one case. The findings highlight the varying effects of mobility restrictions in different municipalities and underscore the importance of considering local dynamics when implementing control measures.

**3.3.3 Isolation.** This study considers isolating completely a municipality by preventing its inhabitants from traveling to other municipalities and not allowing the entry of outsiders. The results show that when this control measure is applied in La Dorada, there is a reduction of 26 dengue cases in the department. However, the opposite effect occurs when this restriction is implemented in Norcasia, Marmato, Chinchiná, and Manizales, leading to an increase of 24, 9, 17, and 3 dengue cases, respectively. Additionally, isolating all five municipalities together results in an increase of 18 in the total number of dengue cases in the department.

## 3.4 Control scenario 2: Confinement

Dengue cases before and after confinement are shown in Table K in S1 Text, revealing that mobility affects each municipality differently. Negative values in the "variation of cases" column indicate a reduction in dengue cases due to mobility restrictions, while positive values indicate an inverse effect. Moreover, Fig 8 presents a color map where the dark green tone signifies an increase in dengue cases when the quarantine is applied, and the red tone indicates an increment in dengue cases when the quarantine is implemented. In specific municipalities such as Manzanares, Salamina, Aguadas, Risaralda, Aranzazu, La Merced, and Pácora, the confinement of the population reduces dengue cases to zero.

Confinement has varying effects on dengue cases in different municipalities. For example, in La Dorada, Riosucio, and Samaná, where a higher percentage of travelers go to areas with a higher incidence of dengue, confinement leads to a decrease in dengue cases. On the other hand, a similar phenomenon occurs in Norcasia, Marmato, Palestina, Chinchiná, Supía, and Viterbo, but in the opposite direction, with the highest percentage of travelers from these three municipalities going to areas with a lower incidence, resulting in an increase in dengue cases with confinement. The percentages of travelers to municipalities with zero, lower, and higher incidence than the municipality of residence can be seen in Table L in S1 Text. Interestingly, these findings are aligned with those obtained by *Conceição et al.* [56] in a study conducted in

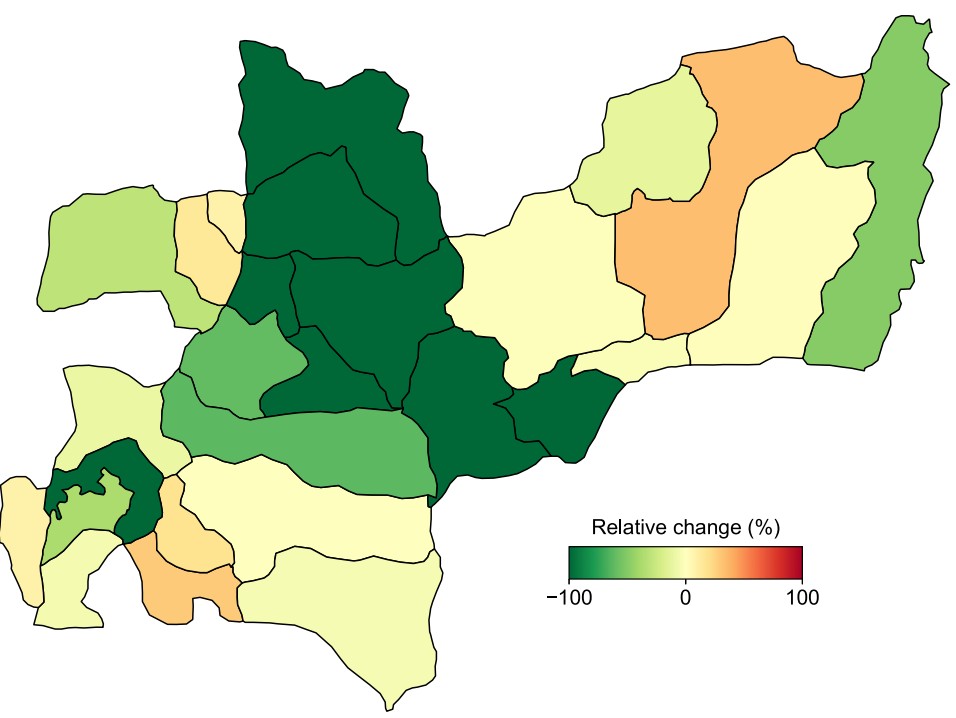

**Fig 8. Variation of dengue cases in the department of Caldas when quarantine are implemented.** The green color indicates a reduction in the number of dengue cases, with darker tones representing more significant changes. Conversely, the red color symbolizes an increment in dengue cases after the quarantine are applied. Map Data Source: The data on this map are the result of simulations through ArcGIS Open Data at Gobernación de Caldas website.

São Paulo (Brasil), where a reduction in dengue cases was observed as a collateral result of population confinement to hinder the advance of the COVID-19 pandemic.

## 4 Discussion

Several scenarios were analyzed to provide policy makers with insights into the implementation of control measures to mitigate the spread of dengue in the department of Caldas. Our study used a compartmental model at each node of a network and investigated the effects of applying mobility control measures on the total number of dengue cases. These measures assume three scenarios, restricting the entrance of individuals to a given municipality, preventing its residents from going to other areas or combining both strategies, thus resulting in a complete isolation of the municipality. The results obtained explain phenomena that have been observed in the historical data, and the coexistence of different types of trends according to local conditions and connectivity.

Our results showed that the isolation of Marmato and Norcasia (a node with a high incidence of the disease) generates an increase in dengue cases. When human movements are limited to one's own home and its surroundings, contact between people and vectors may increase, resulting in a higher risk of exposure and transmission of the virus, as occurred in Thailand where the isolation generated by COVID-19 generated an increase in dengue cases [57]. On the contrary, when residents of Marmato travel to nearby municipalities with a lower incidence of the disease, the probability of contagion is reduced with displacement. In the case of La Dorada, which is also a municipality with a high incidence, the increase in cases is

generated by the departure of its inhabitants. This is due to the fact that most people move to areas with a higher incidence of the disease, which increases the probability of infection and, therefore, the number of infected people. This phenomenon was evidenced in Singapore where the mobility of workers to places with high density of mosquitoes was reduced as a result of COVID-19 quarantine, leading to a decrease in dengue transmission [58]. Therefore, in La Dorada or any municipality with these mobility characteristics, the control measure that generates the greatest impact on the reduction of dengue cases is to limit the exit of its inhabitants.

In this study it was observed that the reduction of dengue cases after applying spatial quarantine is not significantly higher than applying other measures, which calls for caution when implementing these policies since, as observed in the current pandemic, isolation has a negative collateral influence on the population. It causes alterations in eating habits [59], increases the use of drugs [60], has negative psychological [61, 62], and economic [63] effects on people around the world. The analysis carried out in this study can be used, both in the department of Caldas as well as in any other region, to design mobility reduction strategies to control or mitigate a dengue outbreak.

## 5 Conclusion

The model presented in this study makes it possible to analyze the effect of applying control measures to reduce dengue cases in a region. The comparison between the model's predictions (1039 cases) and the actual dengue cases (1033 cases) in the department of Caldas has shown a fair agreement that validates our model as a tool for evaluating different containment policies. To effectively mitigate the spread of dengue during outbreaks, it is recommended that health experts and administrative authorities implement mobility limitations in areas with a high incidence of the disease. Nonetheless, our results indicate that the effectiveness of such policies crucially depends on the infection levels of neighboring areas, observing both positive and negative outcomes of mobility restrictions targeting high incidence patches. The model's applicability can be extended to other regions by incorporating relevant aspects such as average temperature, altitude, population size, human vital statistics, and entomological records. Moreover, the proposed mobility control measures offer valuable insights for informing the development of targeted public health policies in endemic regions. Further research and collaboration are warranted to validate and refine these findings, paving the way for evidence-based strategies to effectively combat dengue transmission.

## Supporting information

**S1 Text. Tables A-L.**
(PDF)

## Acknowledgments

We would like to thank the Health Observatory of Caldas, the Department of Health of Caldas, and the National Coffee Research Center—Cenicafé for granting us access to the database and departmental information for the development of this study.

## Author Contributions

**Conceptualization:** Carolina Ospina-Aguirre, Gustavo Osorio.

**Formal analysis:** Carolina Ospina-Aguirre, David Soriano-Paños, Jesús Gómez-Gardeñes, Gustavo Osorio.

**Investigation:** Carolina Ospina-Aguirre, David Soriano-Paños, Cristian C. Galindo-González, Jesús Gómez-Gardeñes.

**Methodology:** Carolina Ospina-Aguirre, David Soriano-Paños, Jesús Gómez-Gardeñes, Gustavo Osorio.

**Software:** Carolina Ospina-Aguirre, Cristian C. Galindo-González.

**Supervision:** David Soriano-Paños, Gerard Olivar-Tost, Jesús Gómez-Gardeñes, Gustavo Osorio.

**Validation:** Carolina Ospina-Aguirre, David Soriano-Paños, Gerard Olivar-Tost, Jesús Gómez-Gardeñes, Gustavo Osorio.

**Visualization:** Carolina Ospina-Aguirre.

**Writing – original draft:** Carolina Ospina-Aguirre, David Soriano-Paños, Gustavo Osorio.

**Writing – review & editing:** Carolina Ospina-Aguirre, David Soriano-Paños, Gerard Olivar-Tost, Jesús Gómez-Gardeñes, Gustavo Osorio.

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
