## [Decision Letter · Decision Letter 0]

27 Apr 2023

Dear Mrs Ospina,

Thank you very much for submitting your manuscript "Effects of human mobility on the spread of Dengue in the region of Caldas, Colombia." for consideration at PLOS Neglected Tropical Diseases. As with all papers reviewed by the journal, your manuscript was reviewed by members of the editorial board and by several independent reviewers. In light of the reviews (below this email), we would like to invite the resubmission of a significantly-revised version that takes into account the reviewers' comments. 

Dear Author, Thank you so much for your patience. It has been challenging securing peer reviewers but we have finally reach a decision. Pleas address every comments made by each reviewer and provide answers to the their queries.

We cannot make any decision about publication until we have seen the revised manuscript and your response to the reviewers' comments. Your revised manuscript is also likely to be sent to reviewers for further evaluation.

Sincerely,

Mabel Carabali, M.D., M.Sc., Ph.D.,

Academic Editor

Dileepa Ediriweera

Section Editor

Dear Author, Thank you so much for your patience. It has been challenging securing peer reviewers but we have finally reach a decision. Pleas address every comments made by each reviewer and provide answers to the their queries.

Reviewer's Responses to Questions

**Key Review Criteria Required for Acceptance?**

**Methods**

-Are the objectives of the study clearly articulated with a clear testable hypothesis stated?

-Is the study design appropriate to address the stated objectives?

-Is the population clearly described and appropriate for the hypothesis being tested?

-Is the sample size sufficient to ensure adequate power to address the hypothesis being tested?

-Were correct statistical analysis used to support conclusions?

-Are there concerns about ethical or regulatory requirements being met?

Reviewer #1: The study is well structured according to the objectives and hypothesis. Methods are clear and sequential.

Reviewer #2: (No Response)

**Results**

-Does the analysis presented match the analysis plan?

-Are the results clearly and completely presented?

-Are the figures (Tables, Images) of sufficient quality for clarity?

Reviewer #1: Results are according to the proposed methods and are clearly presented. Some figures as stated on the attached comments could be improved to facilitate a wider public comprehension.

Reviewer #2: (No Response)

**Conclusions**

-Are the conclusions supported by the data presented?

-Are the limitations of analysis clearly described?

-Do the authors discuss how these data can be helpful to advance our understanding of the topic under study?

-Is public health relevance addressed?

Reviewer #1: Conclusions are supported by the data. Limitations to the possible applications could be further analysed and discused. The scenarios limiting human movilization as happened in COVID pandemy could be hard to be implemented again.

Reviewer #2: (No Response)

**Editorial and Data Presentation Modifications?**

Reviewer #1: Minor revision

Reviewer #2: (No Response)

**Summary and General Comments**

Reviewer #1: The article provides relevant information for understanding the dynamics of dengue in Colombia, using modeling techniques and suggesting measures to control the spread of the disease taking into account human mobility strategies. Although, the implementation of the strategies could be difficult, there is novelty in the methods and in the use of mobility and disease information. After a minor review, I recommend the article for publication.

Reviewer #2: Please see the attached pdf file for my review.

PLOS authors have the option to publish the peer review history of their article (what does this mean?). If published, this will include your full peer review and any attached files.

Reviewer #1: Yes: Alfredo Acosta

Reviewer #2: No
---

## [Editor Report · Decision Letter 1]

21 Sep 2023

Dear Mrs Ospina,

We are pleased to inform you that your manuscript 'Effects of human mobility on the spread of Dengue in the region of Caldas, Colombia.' has been provisionally accepted for publication in PLOS Neglected Tropical Diseases.

Best regards,

Dileepa Ediriweera

Section Editor

Dileepa Ediriweera

Section Editor

---

## [Editor Report · Acceptance letter]

20 Nov 2023

Dear Mr Soriano-Paños,

We are delighted to inform you that your manuscript, "Effects of human mobility on the spread of Dengue in the region of Caldas, Colombia.," has been formally accepted for publication in PLOS Neglected Tropical Diseases.

Best regards,

Shaden Kamhawi

co-Editor-in-Chief

Paul Brindley

co-Editor-in-Chief
